# Detection and prevalence of a novel *Bandavirus* related to Guertu virus in *Amblyomma gemma* ticks and human populations in Isiolo County, Kenya

Hellen Koka[1,2]*, Solomon Langat[1], Samuel Oyola[3], Faith Cherop[4], Gilbert Rotich[4], James Mutisya[1], Victor Ofula[1], Konongoi Limbaso[1], Juliette R. Ongus[2], Joel Lutomiah[1], Rosemary Sang[4]

1 Kenya Medical Research Institute, Centre for Virus Research, Nairobi, Kenya, 2 Jomo Kenyatta University of Agriculture and Technology, Nairobi, Kenya, 3 International Livestock Research Institute, Nairobi, Kenya, 4 International Centre of Insect Physiology and Ecology, Nairobi, Kenya

* hellenkoka@gmail.com

**Data Availability Statement:** All data generated or analyzed in this study are included in this published article and its supplementary files. Accession

## Abstract

### Introduction

Emerging tick-borne viruses of medical and veterinary importance are increasingly being reported globally. This resurgence emphasizes the need for sustained surveillance to provide insights into tick-borne viral diversity and associated potential public health risks. We report on a virus tentatively designated Kinna virus (KIV) in the family *Phenuiviridae* and genus *Bandavirus*. The virus was isolated from a pool of *Amblyomma gemma* ticks from Kinna in Isiolo County, Kenya. High throughput sequencing of the virus isolate revealed close relatedness to the Guertu virus. The virus genome is consistent with the described genomes of other members of the genus *Bandavirus*, with nucleotides lengths of 6403, 3332 and 1752 in the Large (L), Medium (M) and Small (S) segments respectively. Phylogenetic analysis showed that the virus clustered with Guertu virus although it formed a distinct and well supported branch. The RdRp amino acid sequence had a 93.3% identity to that of Guertu virus, an indication that the virus is possibly novel. Neutralizing antibodies were detected in 125 (38.6%, 95% CI 33.3–44.1%) of the human sera from the communities in this region. In vivo experiments showed that the virus was lethal to mice with death occurring 6–9 days post-infection. The virus infected mammalian cells (Vero cells) but had reduced infectivity in the mosquito cell line (C636) tested.

### Conclusion

Isolation of this novel virus with the potential to cause disease in human and animal populations necessitates the need to evaluate its public health significance and contribution to disease burden in the affected regions. This also points to the need for continuous monitoring of vector and human populations in high-risk ecosystems to update pathogen diversity.

numbers for the L, M and S segments of the virus are PP580190-PP580192 respectively.

**Funding:** This study was partially funded by a grant from Mawazo Institute (https://mawazoinstitute.org), grant number 2022-1-17 and support from the government of Kenya through the Kenya Medical Research Institute. The views expressed herein do not necessarily reflect the official opinion of the donors. The funders had no role in the design of the study; in the collection, analyses, or interpretation of data; in the writing of the manuscript; or in the decision to publish the results.

**Competing interests:** The authors declare that they have no competing interests.

## Introduction

The incidence of tick-borne diseases is increasing globally and has been attributed to multiple factors including climate change that have led to the expansion of tick vectors into new geographical habitats [1]. Tick-borne viruses belong to nine families: *Asfaviridae*, *Flaviviridae*, *Reoviridae*, *Orthomoxyviridae*, *Rhabdoviridae*, *Nyamiviridae*, *Phenuiviridae*, *Nairoviridae* and *Peribunyaviridae* [2] with over 160 viruses that affect humans and animals [3]. Some of the newly identified tick-borne viruses include *Dabie bandavirus* also known as Severe Fever Thrombocytopenia Syndrome Virus (SFTSV) in China, Heart land virus and the Deer tick Virus in North America, Kyasanur Forest Disease in India and Alkhurma Hemorrhagic Fever in Saudi Arabia [4]. The unprecedented number of novel tick-borne viruses being reported, can also be attributed to technological advances in molecular biology [5]. Most of the novel viruses have been placed in established genera although some of the viruses are yet to be assigned to a family or genus [6]. Currently, the genus *Bandavirus* as described in the International Committee on Taxonomy of Viruses (ICTV) 2021 release, consists of eight tick-borne bunyaviruses in the order *Bunyavirales* and family *Phenuiviridae*. These include *Dabie bandavirus*, *Bhanja bandavirus*, *Guertu bandavirus*, *Heartland bandavirus*, *Hunter Island bandavirus*, *Kismaayo bandavirus*, *Lone Star bandavirus* and *Razdan bandavirus* [7]. The representative species are *Dabie bandavirus* (SFTSV), Bhanja virus (BHAV), Guertu virus (GTV), Heartland virus (HRTV), Hunter Island virus (HUIV), Kismaayo Virus (KISV), Lone star virus (LSV) and Razdan virus [8].

   *Dabie bandavirus* (SFTSV), an emerging tick-borne hemorrhagic fever virus, was first isolated in patients from rural China in 2009 [9] and has since been reported in several Asian countries including Japan, South Korea and Vietnam [10–12]. This virus is associated with high fatality rates of 12–50% with clinical symptoms that include fever, myalgia, gastrointestinal symptoms and laboratory abnormalities such as regional lymphadenopathy, thrombocytopenia, leucopenia and elevated serum hepatic enzymes [13]. Severe cases progress to hemorrhagic manifestations, neurologic symptoms followed by multiple organ failure and death, with the most vulnerable being the elderly and immunocompromised individuals [14]. Although nosocomial infections were reported in China through contact with infected blood or mucus, the virus is mainly transmitted by ticks [15, 16]. In China, the virus was detected in *Rhipicephalus microplus* and *Haemaphysalis longicornis* ticks, while transtadial as well as transovarial transmission has been documented [17]. Various wild and domestic animals such as deer, rodents, boars, cattle, goats, pigs, dogs and birds are also implicated in the transmission cycle of this virus with goats and sheep showing high sero-prevalence rates [18, 19]. In the United States, *Amblyomma americanum* have been implicated in the transmission of this virus [20]. Another novel virus, Heartland Virus, that is genetically related to SFTSV was also reported in the United States in 2009, in patients hospitalized with fever, fatigue, anorexia and diarrhea with laboratory abnormalities similar to those of SFTSV [21]. Guertu virus, also a novel highly pathogenic bunyavirus, was first isolated in 2014 from *Demarcentor nuttalli* ticks in the Guertu mountain region of Wusu Xinjiang in China [22, 23]. It is genetically related to the SFTSV and Heartland virus, with up to 90% sequence identity to SFTSV. Guertu virus infects animal and human cell lines and is pathogenic to mice [23]. Antibodies with neutralizing activity against Guertu virus were detected in human serum samples in the Wusu region indicating the viruses' potential to infect humans [23]. Guertu virus, like all *Bandaviruses*, has a tripartite RNA genome consisting of the large (L), Medium (M) and Small (S) segments. The L segment contains one large open reading frame (ORF) which encodes for the RNA-dependent RNA polymerase (RdRp) gene. The M segment also contains a single ORF that encodes for the glycoprotein precursor (comprising Gn and Gc genes), while the S segment contains

two ORFs that encodes for the nucleocapsid (N) and nonstructural proteins (NSs) which exhibits an ambisense gene organization [24]. Several tick-borne viruses including Guertu, were reported in *Hyalomma* ticks from Saudi Arabia [25]. Neutralizing antibodies to Guertu virus were also reported in human samples from Pakistan [26]. Recent reports of Jingmen tick virus in Kenya, underscore the need for continued surveillance of tick-borne viruses and an assessment of the risk they pose to the public health in pastoralist communities [27, 28]. Pastoralist communities live in dry land systems and the livestock interface with wildlife in search of pasture [29] posing a risk for disease transmission [30]. This study describes a virus that was isolated from *Amblyomma gemma* ticks collected from domestic animals in pastoralist communities in Isiolo County, Kenya. Full genome sequence analysis determined the virus to be closely related to Guertu virus and herein we provide data on its phylogeny and sero-prevalence in humans in this dry land ecosystem.

## Materials and methods

### Study site

Isiolo County is an arid and semi-arid area with annual rainfall ranges of 150–600 mm and temperatures range of 24˚C to 30˚C. The land is covered by bushlands, scrubs and shrubs that are utilized as pastoral grazing land and managed by conservancies in the community led by a board of local trustees. Wildlife and tourism are fundamental to this county as it has many wildlife reserves such as Shaba, Buffalo springs, and Bisanadi. On the border of Isiolo and Meru county is Meru national park that is close to some of these villages in this study [31]. Farming is limited to areas around Isiolo central and Kinna that are close to the Ewaso Nyiro River and other tributaries such as the Ngare Mara and Kinna among others.

**Study populations.** Ticks were sampled from four sites in Isiolo County, namely Kinna, Kulamawe, Garbartula and Merti from 2015–2017. Ticks were collected from different domestic animal hosts in September 2015, April 2016 and December 2017. All ticks from one animal per site were stored in 50 ml centrifuge tubes and preserved in liquid nitrogen shippers for transportation to the laboratory. The archived serum samples used in this study were collected in 2015 during a community-based surveillance study targeting healthy individuals from five villages namely:—Kinna, Garfasa, Dadachabasa, Korbesa and Bulesa in Merti and Garbatula districts. The study sought to determine human exposure to arboviruses such as Rift Valley fever virus in nomadic populations along animal migratory routes at points of human-livestock and wildlife convergence (KEMRI SSC No.2346) [32]. The samples are being used retrospectively in this study and were accessed on 11[th] November 2022. The data had been de-identified and thus authors could not identify individual participants during or after data collection. The explanatory variables for the human samples were sex, age, occupation, contact with animals such as cows, goats, donkeys, camels and previous illness that was severe and/or hemorrhagic (Fig 1). This study was approved by the Kenya Medical Research Institute ethics review board protocol number (KEMRI/SERU/CVR/007/4514).

### Laboratory procedures

**Tick identification.** Ticks were identified to species using available morphological keys [33, 34] and pooled (1–8 ticks per pool). Several (10–12) 2 mm Zicornia beads were added to each tick pool in a 1.5 ml centrifuge tube. The centrifuge tubes were frozen before tick homogenization using the Omni bead ruptor-24 at a speed of 3.7 m/s for 1 minute. Homogenizing media prepared by supplementing Minimum Essential Medium (MEM) with 15% Fetal Bovine Serum (FBS), 2% L- glutamine, 2% antibiotic-antimycotic solution (100 units/ml penicillin, 100 µg/ml streptomycin and amphotericin) was added to the tubes of crushed ticks,

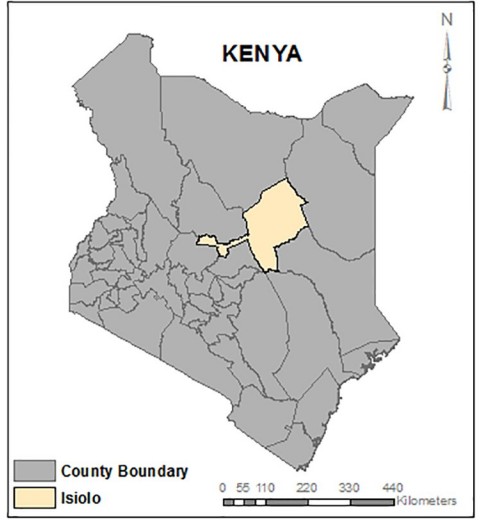

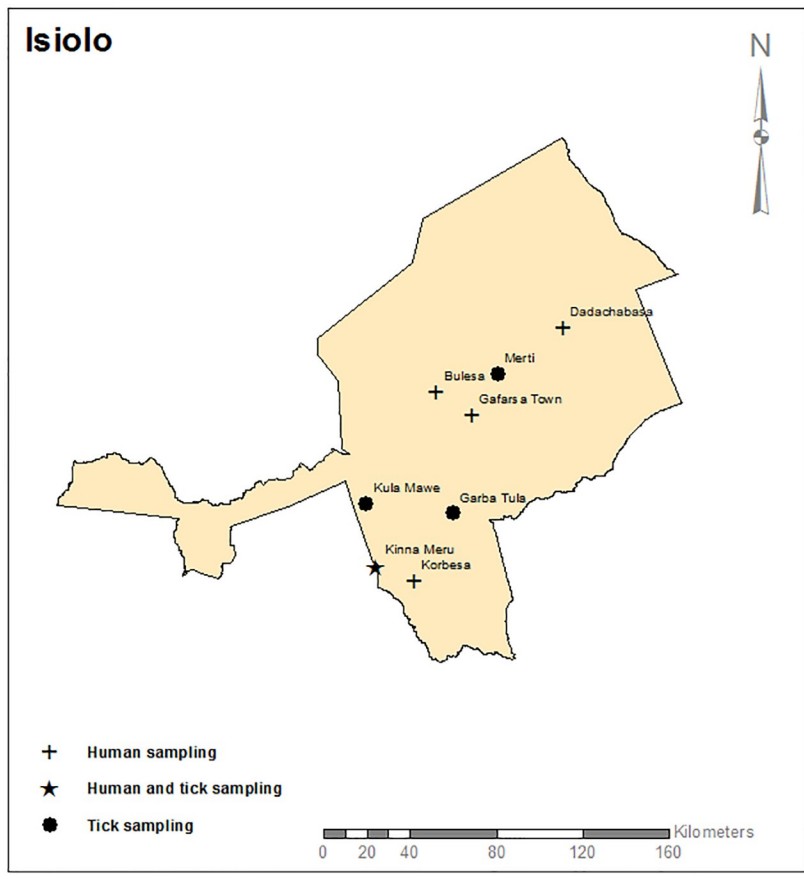

**Fig 1. Map of Isiolo County showing the sites where tick and human samples were collected in this study.** Base maps, boundaries and shape files of Kenyan map and administrative boundaries of the county and Sub-county were derived from GADM data version 4.1 (https://gadm.org) and the maps were generated using ArcGIS Version 10.2.2 (http://desktop.arcgis.com/en/arcmap) advanced license) courtesy of Samuel Owaka.

vortexed, followed by centrifugation at 10,000 rpm for 5 minutes. The supernatants were transferred to cryovials for storage at -80°C until samples were assayed.

**Virus isolation by cell culture and primer design.** Vero CCL-81 (ATTC) cell lines maintained in an incubator at 37°C with 5% $CO_2$ (Sanyo inCu safe, Japan) were used in virus isolation. The cell lines were seeded in 24-well plates and grown to 80–90% confluence in Minimum Essential Medium (MEM) supplemented with 10% Fetal Bovine Serum (FBS), 2% L- glutamine, 2% antibiotic-antimycotic solution (100 units/ml penicillin, 100 μg/ml streptomycin and amphotericin) and 2 ml of Non-Essential Amino Acid (NEAA). After inoculation with the tick supernatant, the cell lines were grown in MEM supplemented with 2% FBS, 2% L-glutamine, 2% antibiotic-antimycotic solution and 2 ml NEAA. Cells were observed daily for viral Cytopathic Effect (CPE) for a maximum of 14 days. Tick supernatants were harvested upon observation of CPE causing 50–70% cell destruction. All the harvested tick pools were passaged to improve chances of virus recovery [35]. The pooled minimum infection rate (Number of positive pooled samples/Total number of ticks tested per 100 ticks) was calculated

for the tick species inoculated [36]. The harvested virus isolates were subjected to an RT-PCR using universal primers targeting flaviviruses, orthobunyaviruses and alphaviruses [37] as well as tick-borne viruses [38]. Virus isolates that could not be identified using available primers were sequenced and new primers were designed from the sequences of the novel virus using the PrimerQuest tool (IDT technologies). PCR products were separated on a 2% agarose gel using Diamond nucleic acid dye (Promega).

**Sequencing and bioinformatics analysis.** The tick supernatant with viral CPE was harvested, centrifuged and then filtered using a 2.0 μm filter to remove cell debris. RNA was extracted from the supernatant using the QIAamp Viral RNA mini kit (Qiagen, Germany), according to the manufacturer's protocol. RNA was quantified using the Qubit 2.0 fluorometer with the Qubit RNA HS assay kit (Invitrogen, USA). RNA libraries were prepared using the TruSeq mRNA Library Prep kit (Illumina, San Diego, CA, USA), following the manufacturer's recommended protocol which was modified to exclude the mRNA clean-up steps [39]. The libraries were sequenced on an Illumina Miseq platform (Illumina, San Diego, CA, USA) using a 2x300 base paired-end reads. Raw sequence reads were inspected for quality using FastQC and subsequently filtered with Prinseq Lite v0.20.4. Cleaned paired reads were assembled *de novo* using Megahit v1.1.2 and the generated contigs were compared to NCBI nucleotide database using a cut-off $E$-value $< 10^{-5}$.

Phylogenetic analysis was performed using MEGA7 [40]. Representative reference genomes were downloaded from GenBank through the NCBI Virus portal and combined with the sequence of the virus generated in this study. Alignment of the combined sequences was achieved with Muscle plugin embedded in MEGA7. Maximum Likelihood phylogenetic analysis was carried out with a bootstrap of 1000 replicates.

**RNA extraction and RT-PCR.** RNA was extracted from 200 tick pool samples from the different sites and the CPE-positive tick isolate was used as a control. Thereafter, the samples were screened for the novel virus using the new primers and superscript III One-step RT-PCR system with platinum taq High Fidelity DNA polymerase according to the manufacturer's instruction (Invitrogen). The following cycling conditions were used with an initial cDNA pre-denaturation of 50˚C for 20 minutes, followed by 94˚C for 2 minutes and 40 cycles of 94˚C for 15 seconds, 62˚C for 30 seconds, 68˚C for 1 minutes and a final extension of 68˚C for 5 minutes. The PCR products were viewed on a 2% gel.

**In vitro virus growth kinetics.** Vero E6, Vero Biken (JCRB0111), Vero CCL-81 and C6/36 cells seeded at $2.8x10^6$ cell/ml were infected with the virus ($5.3 x10^6$ PFU/ml) at a multiplicity of infection (MOI) of 1. After every 24 hours for 7 days, 200 μl of cell culture supernatant was harvested from every infected cell line. Viral titres in these supernatants were quantified by plaque assay and a one-step growth curve generated.

**In vivo studies in mice.** This study was carried out following the recommendations in the Guide for the Care and Use of Laboratory Animals. The protocol was approved by the KEMRI Animal Care and Use Committee (KEMRIACUC/02.07.2023). Three female lactating mice each with a litter of 6–10 Swiss Albino mice (1–2 day-old) per cage from the KEMRI animal house were used in this study to determine virus pathogenicity. A 10-fold dilution of the virus was done and 0.01 ml of the neat virus, $10^{-2}$ and $10^{-4}$ virus dilution were inoculated intracerebrally into the cranium of 1–2-day-old mice in a class II type A2 bio safety cabinet [41]. All the litters of one female were inoculated with the same virus dose. A second replicate of mouse brain inoculations was done with the three virus dilutions but with different batches of litters. A total of 22 suckling mice were used in the first replicate and 24 suckling mice in the second replicate with 6 negative controls for each replicate. Un-infected suckling mice of the same age were used as the negative control group. The virus dilutions of the neat virus, $10^{-2}$ and $10^{-4}$ used in the experimental infection were quantified by plaque assay as follows: - $5.0x10^6$,

$2.7 \times 10^4$ and $1 \times 10^3$ PFU/ml respectively. All animal welfare considerations were taken to minimize suffering or distress. Mice were fed *adlibitum*. All mice were observed twice daily until humane endpoint or the end of the 21-day monitoring period. Specific criteria (humane endpoint) included immobility or paralysis [42]. Any moribund mice were separated, euthanized by cervical dislocation and preserved at -80˚C immediately. Brain samples were harvested and homogenized in 1 ml of cell culture media and a plaque assay test was done to confirm virus infection in the mice.

**Plaque reduction neutralization test.**　The serological assay was done on 324 human sera. The serum samples were heat inactivated at 56˚C for 30 minutes and ten -fold dilutions from 1:20 to 1:320 were tested for neutralizing antibodies by a 90% plaque reduction neutralization test ($PRNT_{90}$) in Vero cells. The virus was diluted to a standard concentration that gave at least 50 plaques. The serum dilutions were mixed with the standard concentration of the diluted virus and incubated for 1 hour at 37˚C. The virus–antibody mixture was inoculated on confluent Vero cells and incubated for 1 hour for virus adsorption. After the adsorption, 2.5% methylcellulose 4000 CV (Sigma) mixed with 2X MEM, was added into the wells [32]. At day 6 post-infection, the plaques were fixed using 3.7% formaldehyde (Sigma) and stained with 0.5% crystal violet (Sigma) in absolute ethanol. Serum samples were considered positive by $PRNT_{90}$ when serum dilution of 1:20 or greater reduced the viral formation of plaques by at least 90% [43].

**Data management.**　All data was entered into an excel database. The sero-prevalence data was imported into Stata version 17.0 for analysis (Stata Corp, College Station, TX). Proportions positive for virus were compared with socio-demographic characteristics using Chi-square test. All tests were performed at 5% significance level.

## Results

### Tick abundance and virus characterization

A total of 2342 ticks were collected and pooled to 409 pools representing 9 species after identification. The most abundant species identified was *Hyalomma marginatum* (40%, 162/409), followed by *Amblyomma gemma* (26%, 105/409), *Rhipicephalus pulchellus* (22.4%, 92/409) and *Hyalomma truncatum* (9%, 38/409). Other species identified included *Rhipicephalus appendiculatus*, *Hyalomma dromedarii*, *Amblyomma variegatum*, *Hyalomma albiparmatum* and *Boophilus* spp. Ticks were predominantly collected from camels (73.1%, 299/409) and cattle (26.2%, 107/409). An isolate from a homogenate of *A. gemma* tick species, induced cytopathic effects between 3–7 days post-infection. The calculated virus pooled minimum infection rate for ticks at this site was *0.04*. The isolate was negative by RT-PCR for flaviviruses, orthobunyaviruses and alphaviruses. It was also negative for Dhori, Thogoto, Dugbe and CCHFV tick-borne viruses. Primer sequences utilized for screening viruses in this study are provided (S1 Table).

### Next generation sequencing

Full genome sequences for the virus indicated that the virus was related to Guertu virus with percent nucleotide identities of approximately 80.42% in the L segment, 76.54% in the M segment and 81.09% in the S segment. The genome architecture was a 3-segmented single stranded RNA and total size of 11847 bp with nucleotides lengths of 6403, 3332 and 1752 in the Large (L), Medium (M) and S (Small) segments respectively. The amino acid identity scores for the different genes encoded by the virus were 93.33% for RdRp, 87.36% in the glycoprotein, 83.86% for the nonstructural protein and 92.24% for the nucleocapsid gene. Phylogenetic analysis placed the isolated virus in the same, but unique cluster with Guertu virus, which further clusters in a similar clade with *Dabie bandavirus* (SFTSV). This observation was consistent across all the four different genes of the virus (Fig 2).

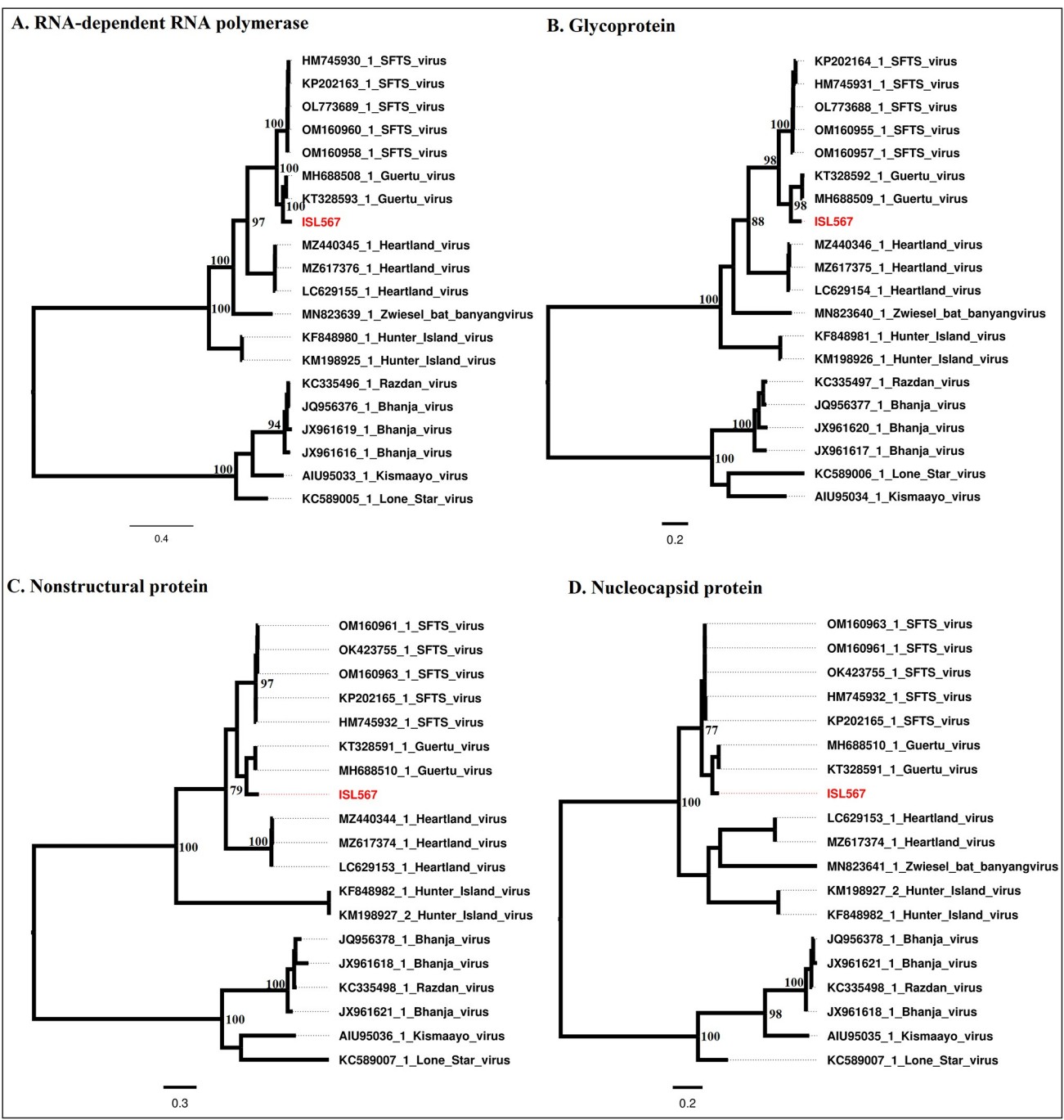

**Fig 2. Phylogenetic analysis of viruses in the genus *Bandavirus*.** The phylogenies were inferred based on RdRp sequences (A), Glycoprotein (B), Nonstructural protein (C) and Nucleoprotein sequences (D). The virus isolated in the study is highlighted in red.

### Prevalence of virus in ticks, cell growth tropism and pathogenicity in mice

The forward and reverse primers with an expected size of 560 bp were designed from the L segment of the virus isolate. The forward- CAAGGCTGAGGATTTGGTATCT and reverse – CTGACTGGGCCCTTTCTATTT sequences for this virus were used to screen more ticks from

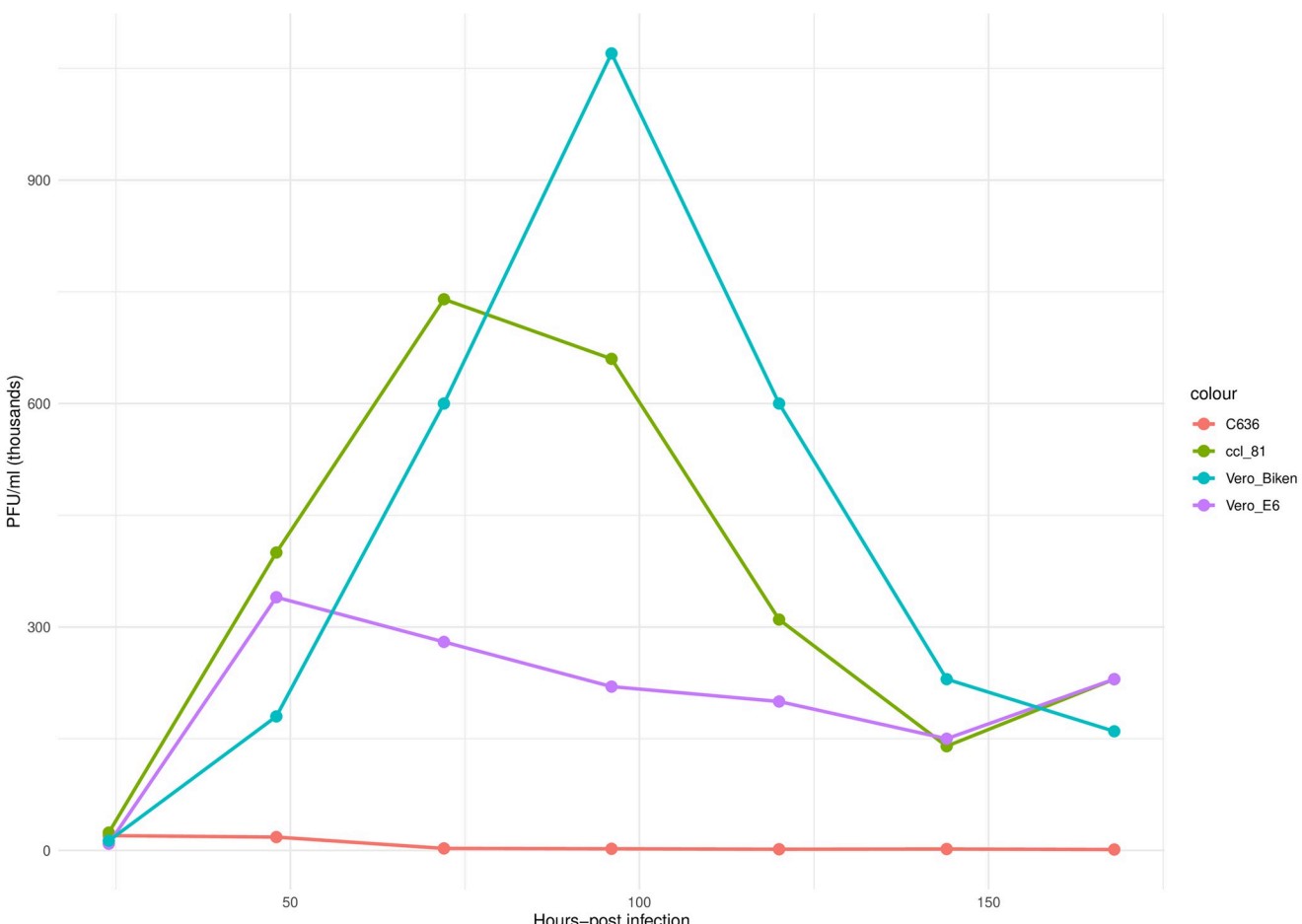

**Fig 3. One-step growth curves of Kinna virus in Vero E6, Vero CCL-81 and C636 at MOI of 1, supernatants were harvested at the indicated time points.**

the region. Nevertheless, the virus was not detected in the additional ticks that were tested. However, in-vitro growth analysis showed that Vero E6, Vero CCL-81 and Vero Biken cells were susceptible to this virus. Peak titres of $1.07 \times 10^6$ PFU/ml were reported in Vero Biken at 72 hours, $7.4 \times 10^5$ PFU/ml at 48 hours for Vero CCL-81 and $3.4 \times 10^5$ PFU/ml at 48 hours for Vero E6 cells. In contrast, virus titres in C636 cells declined after inoculation from $2 \times 10^4$ PFU/ml to $1.3 \times 10^3$ PFU/ml over the 7 days (Fig 3). On the other hand, all the suckling mice (15/15) infected with the undiluted virus were euthanized within 6–9 days post-infection. The mice exhibited signs of sickness including being moribund, difficulty in breathing, decreased growth, loose skin and uncoordinated movement. Two of the mice were found cannibalized by the dam on day 7, probably due to onset of disease. Of the mice infected with $10^{-2}$ virus dilution, (10/14) were euthanized between 8–10 days post infection and two were cannibalized by the dam on day 8 and day 13. Two of the mice in the first group survived 21 days post-infection while two in the second replicate survived to day 13 and were euthanized accordingly. However, none of the suckling mice (7/7) infected with the $10^{-4}$ virus dilution in the first replicate died and were euthanized on day 21. Between 8–10 days post-infection, the mice appeared to be inactive and sickly with ruffled fur but seemed to recover after day 11. In the second replicate, the suckling mice (6/6) infected with the $10^{-4}$ virus dilution, were cannibalized by the

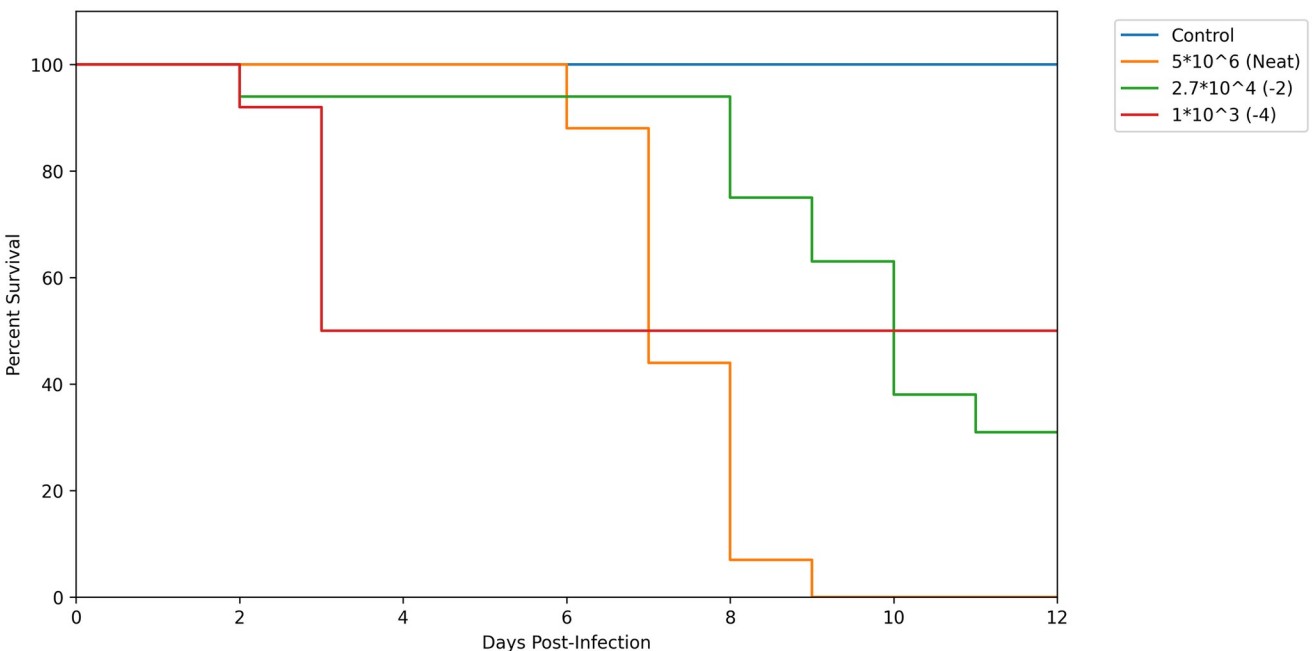

**Fig 4. Survival curves of 1–2 day old Swiss Albino suckling mice inoculated intracerebrally with neat, $10^{-2}$ and $10^{-4}$ virus dilutions.**

dam between days 1–3 post-infections (Fig 4). Mouse brain samples that were harvested were quantified by plaques and titres ranged from $1.1 \times 10^6$ PFU/ml from mice brains inoculated with neat virus to as low as $1 \times 10^4$ PFU/ml for mice brains inoculated with $10^{-2}$ diluted virus.

## Sero-prevalence in humans

The plaque reduction neutralization test was carried out on 324 human samples. Participants were drawn from Garbatula (n = 158) and Merti (n = 166) sub-counties in Isiolo and their age varied from 14–90 years. Antibodies against the virus were detected in 125 (38.6%, 95% CI 33.3–44.1%) of the samples tested. The $PRNT_{90}$ titres ranged from 125 for the virus at 1:20 dilution, 25 (1:40), 17(1:80), 10(1:160) and 3 (1:320). Notably, all the neutralizing titres at 1:320 were detected in serum samples from women aged 25, 60 and 70 years, the former two being farmers and the latter a housewife. The sero-prevalence of the virus was significantly higher in Garbatula (51.9%) than in Merti (25.9%; p<0.001). Most of the sero-positive samples were from people who had resided in the villages for most of their lives. The sero-positivity was significantly higher in those who had contact with cattle (p = 0.011) (Table 1). The sero-positivity also increased with age (Table 2) with a majority of those positive at dilutions 1:80 or higher being in the 55–90 years age bracket.

## Discussion

We report the isolation of a virus in the family *Phenuiviridae* and genus *Bandavirus*, tentatively named Kinna virus that is closely related to Guertu virus from a pool of *A. gemma* ticks from Isiolo County. The ICTV has various criteria for the classification of virus species and this differs depending on the virus group but includes data on genomic, biological and antigenic properties among other information [44]. The species and genus demarcation criteria for *Bandaviruses* requires novel species to have less than 95% sequence identity in the RdRp amino

**Table 1. Virus prevalence by socio-demographic characteristics.**

| Characteristic | Number of participants | Number positive | % positive (95% CI) | Chi square p-value |
|---|---|---|---|---|
| Overall | 324 | 125 | 38.6 (33.3–44.1) | |
| Sub county | | | | <0.001 |
| Garbatula | 158 | 82 | 51.9 (43.8–59.9) | |
| Merti | 166 | 43 | 25.9 (19.4–33.3) | |
| Village | | | | <0.001 |
| Garfasa | 68 | 40 | 58.8 (46.2–70.6) | |
| Kinna | 90 | 42 | 46.7 (36.1–57.5) | |
| Bulesa | 30 | 12 | 40.0 (22.7–59.4) | |
| Dadachabasa | 74 | 20 | 27.0 (17.4–38.6) | |
| Korbesa | 62 | 11 | 17.7 (9.2–29.5) | |
| Sex | | | | 0.337 |
| Female | 210 | 77 | 36.7 (30.1–43.6) | |
| Male | 114 | 48 | 42.1 (32.9–51.7) | |
| Age group in years | | | | 0.313 |
| 14–24 | 39 | 14 | 35.9 (21.2–52.8) | |
| 25–34 | 68 | 22 | 32.4 (21.5–44.8) | |
| 35–44 | 48 | 16 | 33.3 (20.4–48.4) | |
| 45–54 | 55 | 20 | 36.4 (23.8–50.4) | |
| 55–64 | 42 | 17 | 40.5 (25.6–56.7) | |
| 65–90 | 72 | 36 | 50.0 (38.0–62.0) | |
| Occupation | | | | 0.552 |
| Farmer | 201 | 83 | 41.3 (34.4–48.4) | |
| Herdsman | 95 | 31 | 32.6 (23.4–43.0) | |
| Housewife | 16 | 6 | 37.5 (15.2–64.6) | |
| Other | 12 | 5 | 41.7 (15.2–72.3) | |
| Contact with chicken | | | | 0.070 |
| No | 187 | 80 | 42.8 (35.6–50.2) | |
| Yes | 137 | 45 | 32.8 (25.1–41.4) | |
| Contact with goats | | | | 0.807 |
| No | 46 | 17 | 37.0 (23.2–52.5) | |
| Yes | 278 | 108 | 38.8 (33.1–44.9) | |
| Contact with cows | | | | 0.011 |
| No | 105 | 51 | 48.6 (38.7–58.5) | |
| Yes | 219 | 74 | 33.8 (27.6–40.5) | |
| Contact with donkeys | | | | 0.964 |
| No | 251 | 97 | 38.6 (32.6–45.0) | |
| Yes | 73 | 28 | 38.4 (27.2–50.5) | |
| Contact with camels | | | | 0.851 |
| No | 321 | 123 | 38.6 (33.3–44.2) | |
| Yes | 3 | 1 | 33.3 (1.0–90.6) | |
| Had previous severe illness with blood symptoms | | | | 0.315 |
| No | 321 | 123 | 38.3 (33–43.9) | |
| Yes | 3 | 2 | 66.7 (9.4–99.2) | |

acid sequence [45]. Furthermore, based on revisions by Fauquet and Stanley, (2005) [46], tick-borne virus contigs showing greater than 10% difference were considered putative novel species while those with less than 10% difference were identified as strains of known viruses [47]. Since the virus isolate had a 93.3% RdRp amino acid sequence identity to Guertu virus while

**Table 2. Number of samples sero-positive for virus at different dilutions and ages.**

| PRNT result | Age in years | | | | | | |
|---|---|---|---|---|---|---|---|
| | **14–24** | **25–34** | **35–44** | **45–54** | **55–64** | **65–90** | **Total** |
| Negative | 25 | 46 | 32 | 35 | 25 | 36 | 199 |
| 1:20 | 14 | 22 | 16 | 20 | 17 | 36 | 125 |
| 1:40 | 0 | 4 | 0 | 3 | 6 | 12 | 25 |
| 1:80 | 0 | 3 | 0 | 2 | 4 | 8 | 17 |
| 1:160 | 0 | 2 | 0 | 0 | 2 | 6 | 10 |
| 1:320 | 0 | 1 | 0 | 0 | 1 | 1 | 3 |

the contigs showed nucleotide similarity (<90%) to Guertu virus, we postulate that this could be a putative novel species.

A high sero-prevalence was reported in the human samples tested against this virus suggesting previous exposure and infection to this virus. Similar to SFTSV, age was a risk factor as those >55years of age reported higher antibodies titres [48]. Significant risk factors included sub-county of residence and contact with cattle, with farmers and herdsmen being more predisposed to infection. Herdsmen in particular were also shown to be greatly exposed to Guertu virus than farmers in the Xinjiang province, where the virus was first isolated [23]. Although, the sero-prevalence in our study was not significantly different in males and females, the highest dilution that neutralized this virus was reported in samples collected from women. It is speculated that women spend more time in caring for sick animals than their male counterparts, increasing the risk of exposure to diseases [49]. We also attributed exposure to high-risk activities such as milking, slaughtering, skinning of dead animals or handling aborting fetuses. Exposure may also have occurred inadvertently from the bite of an infected tick as the domestic animals were infested. Moreover, there could be a possible zoonotic risk associated with this virus due to the fact that the tick vectors reported in this study also prefer wild hosts [50]. The probability that this virus may have originated from wild animals is high since the villages in this study area are close to several game reserves that formed zones of human-livestock wildlife convergence.

Mice infected intracerebrally with a high dose of the novel virus did not survive. However, survival rates were high in mice infected with the low dilution of virus suggesting that a lower infectious dose might not be detrimental but may aid in maintaining transmission and thus these animals could act as reservoirs of infection. The dose-dependent effect was demonstrated in a study of mice infected with SFTSV that indicated the severity of infection was a factor of the viral load and that a low amount of virus could result in asymptomatic infection [51]. The mice in this study showed signs of clinical disease to Kinna virus and this suggests that the virus could cause disease in humans. The clinical presentation of Guertu virus in humans is not well documented but is believed to be similar to that of *Dabie bandavirus* [23]. We hypothesize that the clinical presentation of Kinna virus could be similar to that of Guertu virus as the two viruses share many characteristics including tick vector, ability to infect different cell lines and pathogenicity in mice. Since the symptoms of the related virus, *Dabie bandavirus* are known, we recommend that clinicians in this region look out for this clinical syndrome.

Previous studies have reported Isiolo county to be a high-risk area for zoonotic diseases such as Rift Valley fever virus, yellow fever virus and *Brucella* spp. [52, 53]. These diseases share the same non-specific clinical syndrome with many tick-borne viruses which can lead to misdiagnosis. Thus, the identification of this novel virus from ticks in this county is evidence of the existence of uncharacterized viruses at the human-livestock convergence zones which

may be responsible for the burden of undiagnosed febrile illness reported during outbreaks. Our findings, thus underscore the need for increased surveillance to understand the pathology of this new virus, susceptible hosts and reservoirs, and development of appropriate point of care diagnostic tools.

This research is subject to some limitations. The novel virus was isolated from one pool of *A. gemma* ticks in this study. This is not unusual in vector studies as virus detection rates may be very low in the absence of outbreaks or during inter-epidemic periods thus the need to test many samples in order to increase chances of virus isolation [54]. Although, the tick samples tested were not collected from the exact locations as the human samples, the pastoralist system in this region, ensures the movement of animals and by extension the tick vectors which may facilitate transmission of infectious pathogens to human populations. The serological data reported indicates that the novel virus is circulating in this region, nevertheless, no other B*andaviruses* have been reported locally thus cross-neutralization data could not be generated. Additionally, the participants of the sero-survey were healthy individuals, therefore symptoms of the infection could not be determined and thus further studies to identify the clinical manifestation associated with this novel virus may need to be undertaken. Lastly, animal hosts were not sampled due to logistical limitations. The study however, recommends incorporation in future studies to generate evidence of exposure or active infection and thus understand the implication of this virus on animal health.

## Conclusions

We report the isolation of a novel *Bandavirus* from Isiolo County with a potential to cause disease in human and animal populations. A one-health approach should be considered in the evaluation of the public health significance of this virus and for development of appropriate control strategies. Detection of this new strain also points to the need for continuous monitoring of vector and human populations in high-risk ecosystems in order to provide an update on pathogen diversity.

## Supporting information

**S1 Table. A list of conventional PCR primers used in this study.**
(DOCX)

## Acknowledgments

I thank Reuben Lugalia, John Gachoya, Francis Mulwa and Dunstone Beti for their participation in the collection and identification of the ticks used in this study. We are also grateful to Samuel Owaka and Bryson Kimemia for the graphics in this publication.

## Author Contributions

**Conceptualization:** Hellen Koka, Juliette R. Ongus, Joel Lutomiah, Rosemary Sang.

**Data curation:** Hellen Koka, Solomon Langat, Samuel Oyola.

**Formal analysis:** Hellen Koka, Solomon Langat, Samuel Oyola.

**Investigation:** Hellen Koka, Faith Cherop, Gilbert Rotich, James Mutisya, Victor Ofula, Konongoi Limbaso.

**Methodology:** Hellen Koka, Faith Cherop, Gilbert Rotich, James Mutisya, Victor Ofula, Konongoi Limbaso, Juliette R. Ongus, Joel Lutomiah, Rosemary Sang.

**Resources:** Juliette R. Ongus, Joel Lutomiah, Rosemary Sang.

**Software:** Solomon Langat, Samuel Oyola.

**Supervision:** Juliette R. Ongus, Joel Lutomiah, Rosemary Sang.

**Validation:** Hellen Koka, Solomon Langat, Samuel Oyola, Rosemary Sang.

**Visualization:** Hellen Koka, Solomon Langat.

**Writing – original draft:** Hellen Koka.

**Writing – review & editing:** Hellen Koka, Solomon Langat, Samuel Oyola, Faith Cherop, Gilbert Rotich, James Mutisya, Victor Ofula, Konongoi Limbaso, Juliette R. Ongus, Joel Lutomiah, Rosemary Sang.

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
