## [Decision Letter · Decision Letter 0]

24 Jun 2024

PONE-D-24-20220

Detection and prevalence of a novel Bandavirus related to Guertu virus in Amblyomma gemma ticks and human populations in Isiolo County, Kenya

PLOS ONE

Dear Dr.  Koka,

Thank you for submitting your manuscript to PLOS ONE. After careful consideration, we feel that it has merit but does not fully meet PLOS ONE’s publication criteria as it currently stands. Therefore, we invite you to submit a revised version of the manuscript that addresses the points raised during the review process.

We look forward to receiving your revised manuscript.

Kind regards,

Daniel Oladimeji Oluwayelu, D.V.M., Ph.D.

Academic Editor

PLOS ONE

Journal Requirements:

5. We note that Figure 1 in your submission contain map/satellite images which may be copyrighted. All PLOS content is published under the Creative Commons Attribution License (CC BY 4.0), which means that the manuscript, images, and Supporting Information files will be freely available online, and any third party is permitted to access, download, copy, distribute, and use these materials in any way, even commercially, with proper attribution. For these reasons, we cannot publish previously copyrighted maps or satellite images created using proprietary data, such as Google software (Google Maps, Street View, and Earth). For more information, see our copyright guidelines: http://journals.plos.org/plosone/s/licenses-and-copyright.

Reviewers' comments:

Reviewer's Responses to Questions

**Comments to the Author**

1. Is the manuscript technically sound, and do the data support the conclusions?

Reviewer #1: Yes

Reviewer #2: Yes

2. Has the statistical analysis been performed appropriately and rigorously? 

Reviewer #1: Yes

Reviewer #2: Yes

3. Have the authors made all data underlying the findings in their manuscript fully available?

Reviewer #1: Yes

Reviewer #2: Yes

4. Is the manuscript presented in an intelligible fashion and written in standard English?

Reviewer #1: Yes

Reviewer #2: Yes

5. Review Comments to the Author

Reviewer #1: Review of ‘Detection and prevalence of a novel Bandavirus related to Guertu virus in Amblyomma gemma ticks and human populations in Isiolo County, Kenya’

The study looked at the detection of a novel virus belonging to the Bandavirus genus in Amblyomma ticks and humans in Kenya. The study is interesting. The study however does not include its limitations.

I noted following limitations in the study. Although a large number of ticks were tested, only in one pool the novel virus was detected, whereas the in human population more than 38% has serological evidence of the virus. I found that the cut-off titre for seropositives was quite low. According to me, I would have only considered titres from 1:160. Little is also discussed about cross-reactivity with other viruses and false positives. I do not know what is the sensitivity and specificity of the plague reduction virus neutralization test. This information should have been included in the methods and limitations should have been discussed at the end of the manuscript. In addition, the study would have had more power if the samples from ticks came animals that were owned/herded by the humans that were samples. As far as I understood the samples were from same counties but tick and human samples were not linked to same location. In addition, it would have provided more insight if also animals were sampled. This should have been included as limitations of the study. Suggestion are made that the novel virus can cause the same symptoms of humans as the related Guertu virus but yet it seems although evidence of past infection was find in a high number of humans but yet those people were healthy at the time of sample. Although the sample size is small, I find it odd that the conclusion was made that the virus could cause serious disease in humans yet there was no evidence because people were health although had been infected previously based on serological evidence. The route of exposure is also not well understood yet. Are humans exposed through adult Amblyomma ticks from accidental close contact with cows or are do they acquire infection from the immature stages directly from environment. For example African tick bite fever is transmitted more likely from the immature stages of Amblyomma ticks.

I think above issues and limitations could be added and discussed in the paper.

For the remainder, I think this is a very interesting paper.

Reviewer #2: This study isolates and characterizes a type of novel Bandavirus. Experiments to determine its pathogenicity as well as prevalence in humans were performed. The virus could have potentially important health consequences in humans. Authors perform rigorous experimental methods and conclude that this virus is closely related to Guertu virus within the genus. This is an important paper that needs to be published. I have added minor comments in a marked up text.

6. PLOS authors have the option to publish the peer review history of their article (what does this mean?). If published, this will include your full peer review and any attached files.

Reviewer #1: No

Reviewer #2: No

---

## [Author Response · Author response to Decision Letter 0]

24 Jul 2024

PONE-D -24-20220

Dear Editor,

Thank you for the review comments. Please find below responses to the reviewers and editors comments.

Regards,

Hellen Koka

RESPONSES TO REVIEWER #1 COMMENTS

1. Review of ‘Detection and prevalence of a novel Bandavirus related to Guertu virus in Amblyomma gemma ticks and human populations in Isiolo County, Kenya’. The study looked at the detection of a novel virus belonging to the Bandavirus genus in Amblyomma ticks and humans in Kenya. The study is interesting. The study however does not include its limitations. I noted following limitations in the study. Although a large number of ticks were tested, only in one pool the novel virus was detected, whereas the in human population more than 38% has serological evidence of the virus. 

Response: That we found one pool testing positive for the virus at the time, for us was exciting. The number of detections of a virus often depends on its level of circulation at the point of sampling, which is influenced by factors that are not known to us at this point in time, Line 363-366. Detection of one positive pool for us is a very important finding and evidence of circulation of this virus in our region. Serologic evidence of circulation represents possible cumulative exposure of the population over time, an important additional finding. This finding opens opportunity for further in-depth surveillance when funding is available. We also tested an additional pool of 200 tick from this region by PCR but did not detect the novel virus again, Line 184 and Line 267.

2. I found that the cut-off titre for sero-positives was quite low. According to me, I would have only considered titres from 1:160. Little is also discussed about cross-reactivity with other viruses and false positives. I do not know what is the sensitivity and specificity of the plaque reduction virus neutralization test. This information should have been included in the methods and limitations should have been discussed at the end of the manuscript. 

Response: The cut-off titre for sero-positivity of this novel virus was retained at 1:20 and since the samples were not tested against other viruses in the Bandavirus genus, no cross-reactivity was reported. There are no known circulating Bandaviruses locally that we could consider for cross-neutralization, Line367-368. The sensitivity and specificity are above 70% and is the reason plaque reduction test is considered a gold standard. 

3. In addition, the study would have had more power if the samples from ticks came from animals that were owned/herded by the humans that were samples. As far as I understood the samples were from same counties but tick and human samples were not linked to same location. 

Response: This suggested sampling plan would have been ideal. The limitation is addressed, Line 366-369. It was after the virus was detected, isolated and characterized that the serologic analysis was considered. For this reason, the human samples were not linked directly to the exact location/herd where the tick samples had been collected. However, some of the human samples were collected from the general locality where the ticks were collected, Fig 1, Line 118 and Line 124. Further, the target communities are largely nomadic and often move through wide areas in search of pasture and water creating potential for movement of animals and ticks over a wider space. This pastoral system thus increases the potential for pathogen transmission in the region. This is also supported by the fact that; the human samples had been collected along the animal migratory routes used by the pastoralists in this region. 

4. In addition, it would have provided more insight if also animals were sampled. This should have been included as limitations of the study.

Response: Due to the scope of the research and level of funding for the work, no animals were sampled, and this has been captured as a limitation of the study. This initial funding opens opportunity for seeking more funding for such additional studies as suggested by the reviewers, Line 374-377.

5. Suggestion are made that the novel virus can cause the same symptoms of humans as the related Guertu virus but yet it seems although evidence of past infection was found in a high number of humans but yet those people were healthy at the time of sample. 

Response: The symptoms caused by this virus are not known. There is need for further studies to determine symptoms associated with acute infection of this virus. At this time, we can only speculate that this novel virus may cause similar course of infection as Guertu or Dabie bandavirus (SFTSV), Line 371-374. The suggestion has been made to promote awareness among clinicians in our region to enhance the index of suspicion as they review patients, Line 352-353.

6. Although the sample size is small, I find it odd that the conclusion was made that the virus could cause serious disease in humans yet there was no evidence because people were healthy although had been infected previously based on serological evidence. 

Response: Please refer to the response in point 5. Although, the symptoms are not known, the virus was also shown to infect mice and thus we speculate that the virus may have potential to cause disease in humans, Line 346-348.

7. The route of exposure is also not well understood yet. Are humans exposed through adult Amblyomma ticks from accidental close contact with cows or are do they acquire infection from the immature stages directly from environment. For example, African tick bite fever is transmitted more likely from the immature stages of Amblyomma ticks.

I think above issues and limitations could be added and discussed in the paper.

Response: The route of exposure is not well understood, it could be through the bite of the tick or through handling of infectious materials from infected animal, Line 334-338.

8. For the remainder, I think this is a very interesting paper.

Response: We appreciate the review comments and the concerns raised which we have used to improve the manuscript. Thank you.

RESPONSE TO REVIEWER #2 COMMENTS

1. This study isolates and characterizes a type of novel Bandavirus. Experiments to determine its pathogenicity as well as prevalence in humans were performed. The virus could have potentially important health consequences in humans. Authors perform rigorous experimental methods and conclude that this virus is closely related to Guertu virus within the genus. This is an important paper that needs to be published. 

I have added minor comments in a marked-up text.

Response: We thank reviewer number 2 for the specific comments on the document that were very valuable in improving the manuscript. We have done our best to respond to all of them. Thank you. 

RESPONSES TO EDITORS COMMENTS

Response: These guidelines have been met

Response: Additional information has been provided for experiments involving animals. Line 214-217

3. When completing the data availability statement of the submission form, you indicated that you will make your data available on acceptance. We strongly recommend all authors decide on a data sharing plan before acceptance, as the process can be lengthy and hold up publication timelines. Please note that, though access restrictions are acceptable now, your entire data will need to be made freely accessible if your manuscript is accepted for publication. This policy applies to all data except where public deposition would breach compliance with the protocol approved by your research ethics board. If you are unable to adhere to our open data policy, please kindly revise your statement to explain your reasoning and we will seek the editor's input on an exemption. Please be assured that, once you have provided your new statement, the assessment of your exemption will not hold up the peer review process

Response: Data sharing plan revised

Response: Ethics statement was added, Line 131-133.

5. We note that Figure 1 in your submission contain map/satellite images which may be copyrighted. All PLOS content is published under the Creative Commons Attribution License (CC BY 4.0), which means that the manuscript, images, and Supporting Information files will be freely available online, and any third party is permitted to access, download, copy, distribute, and use these materials in any way, even commercially, with proper attribution. For these reasons, we cannot publish previously copyrighted maps or satellite images created using proprietary data, such as Google software (Google Maps, Street View, and Earth). For more information, see our copyright guidelines: http://journals.plos.org/plosone/s/licenses-and-copyright.

Response: The map was created from the ARC GIS software by Samuel Owaka and details are provided under the figure caption for Fig 1.

6. Please review your reference list to ensure that it is complete and correct. If you have cited papers that have been retracted, please include the rationale for doing so in the manuscript text, or remove these references and replace them with relevant current references. Any changes to the reference list should be mentioned in the rebuttal letter that accompanies your revised manuscript. If you need to cite a retracted article, indicate the article’s retracted status in the References list and also include a citation and full reference for the retraction 

Response: The references were formatted and checked for completeness. One reference was added, Line 482.

---

## [Decision Letter · Decision Letter 1]

19 Aug 2024

PONE-D-24-20220R1

Detection and prevalence of a novel Bandavirus related to Guertu virus in Amblyomma gemma ticks and human populations in Isiolo County, KenyaPLOS ONE

Dear Dr. Koka,

Thank you for submitting your manuscript to PLOS ONE. After careful consideration, we feel that it has merit but does not fully meet PLOS ONE’s publication criteria as it currently stands. Therefore, we invite you to submit a revised version of the manuscript that addresses the points raised during the review process.

We look forward to receiving your revised manuscript.

Kind regards,

Daniel Oladimeji Oluwayelu, D.V.M., M.Sc., Ph.D.

Academic Editor

PLOS ONE

Journal Requirements:

Additional Editor Comments:

Authors should address the issue raised by Reviewer #1 in Section 6. Review Comments to the Author (below).

Reviewers' comments:

Reviewer's Responses to Questions

**Comments to the Author**

1. If the authors have adequately addressed your comments raised in a previous round of review and you feel that this manuscript is now acceptable for publication, you may indicate that here to bypass the “Comments to the Author” section, enter your conflict of interest statement in the “Confidential to Editor” section, and submit your "Accept" recommendation.

Reviewer #1: All comments have been addressed

Reviewer #3: All comments have been addressed

2. Is the manuscript technically sound, and do the data support the conclusions?

Reviewer #1: Yes

Reviewer #3: Yes

3. Has the statistical analysis been performed appropriately and rigorously? 

Reviewer #1: Yes

Reviewer #3: Yes

4. Have the authors made all data underlying the findings in their manuscript fully available?

Reviewer #1: Yes

Reviewer #3: Yes

5. Is the manuscript presented in an intelligible fashion and written in standard English?

Reviewer #1: Yes

Reviewer #3: Yes

6. Review Comments to the Author

Reviewer #1: Thanks for addressing the comments I had diligently and adding a limitations section.

I have just one small comment re: line 305-307, I recommend to use seroprevalence/seropositivity instead of prevalence because the humans were tested by serology.

Thanks for this interesting study. I accepted the revised manuscript for publication.

Reviewer #3: I reviewed the revisions and noted that all of them have been addressed satisfactorily. The paper presents novel information required to be published.

7. PLOS authors have the option to publish the peer review history of their article (what does this mean?). If published, this will include your full peer review and any attached files.

Reviewer #1: No

Reviewer #3: **Yes: **Samoel Ashimosi Khamadi

---

## [Author Response · Author response to Decision Letter 1]

30 Aug 2024

Editor in Chief

PLOS One

26th August 2024.

Thank you for the review of the manuscript entitled: “Detection and prevalence of a novel Bandavirus related to Guertu virus in Amblyomma gemma ticks and human populations in Isiolo County, Kenya.” The issues raised by the editor has been addressed and below is the responses to the comments.

RESPONSES TO EDITORS COMMENTS

1. We note that Figure 1 in your submission contain map/satellite images which may be copyrighted. All PLOS content is published under the Creative Commons Attribution License (CC BY 4.0), which means that the manuscript, images, and Supporting Information files will be freely available online, and any third party is permitted to access, download, copy, distribute, and use these materials in any way, even commercially, with proper attribution. For these reasons, we cannot publish previously copyrighted maps or satellite images created using proprietary data, such as Google software (Google Maps, Street View, and Earth). For more information, see our copyright guidelines: http://journals.plos.org/plosone/s/licenses-and-copyright.

Response: Thank you for this comment. The maps were not copyrighted. I have edited the caption to Figure 1 as follows;

These maps were generated courtesy of Samuel Owaka using ArcGIS Version 10.2.2 (http://desktop.arcgis.com/en/arcmap) advanced license.

---

## [Editor Report · Decision Letter 2]

4 Sep 2024

PONE-D-24-20220R2

Detection and prevalence of a novel Bandavirus related to Guertu virus in Amblyomma gemma ticks and human populations in Isiolo County, Kenya

PLOS ONE

Dear Dr. Koka,

Thank you for submitting your manuscript to PLOS ONE. After careful consideration, we feel that it has merit but does not fully meet PLOS ONE’s publication criteria as it currently stands. Therefore, we invite you to submit a revised version of the manuscript that addresses the points raised during the review process.

**ACADEMIC EDITOR:** Kindly address the issue raised by Reviewer 1 concerning the use of the word "prevalence" on Lines 305 - 307 of the revised manuscript: "I recommend to use seroprevalence/seropositivity instead of prevalence because the humans were tested by serology". I recommend that you replace "prevalence" with "seroprevalence" on Line 305 and in the title of Table 1.

We look forward to receiving your revised manuscript.

Kind regards,

Daniel Oladimeji Oluwayelu, D.V.M., M.Sc., Ph.D.

Academic Editor

PLOS ONE
---

## [Author Response · Author response to Decision Letter 2]

5 Sep 2024

5th September 2024

Editor in Chief

PLOS One

Thank you for the review of the manuscript entitled: “Detection and prevalence of a novel Bandavirus related to Guertu virus in Amblyomma gemma ticks and human populations in Isiolo County, Kenya.” The issues raised by the editor have been addressed and below is the responses to comments.

RESPONSES TO REVIEWER #1 COMMENTS

1. Thanks for addressing the comments I had diligently and adding a limitations section. I have just one small comment re: line 305-307, I recommend to use sero-prevalence/sero-positivity instead of prevalence because the humans were tested by serology.

Thanks for this interesting study. I accepted the revised manuscript for publication.

Response: The revision of the word prevalence to sero-prevalence / sero-positivity was done, Line 305-307. We appreciate your review comments as they greatly improved this manuscript. Thank you.

RESPONSES TO EDITORS COMMENTS

Response: The references were checked for completeness

---

## [Editor Report · Decision Letter 3]

8 Sep 2024

Detection and prevalence of a novel Bandavirus related to Guertu virus in Amblyomma gemma ticks and human populations in Isiolo County, Kenya

PONE-D-24-20220R3

Dear Dr. Hellen Koka,

We’re pleased to inform you that your manuscript has been judged scientifically suitable for publication and will be formally accepted for publication once it meets all outstanding technical requirements.

Kind regards,

Daniel Oladimeji Oluwayelu, D.V.M., M.Sc., Ph.D.

Academic Editor

PLOS ONE
---

## [Editor Report · Acceptance letter]

12 Sep 2024

PONE-D-24-20220R3 

PLOS ONE

Dear Dr. Koka, 

I'm pleased to inform you that your manuscript has been deemed suitable for publication in PLOS ONE. Congratulations! Your manuscript is now being handed over to our production team.

Kind regards, 

on behalf of

Professor Daniel Oladimeji Oluwayelu 

Academic Editor

PLOS ONE